# Primary care physician perspectives on screening for axial spondyloarthritis: A qualitative study

Kate L. Lapane[1], Divya Shridharmurthy[1,2], Sara Khan[1], Daniel Lindstrom[3], Ariel Beccia[1,2], Esther Yi[4], Jonathan Kay[1,5,6], Catherine Dube[1], Shao-Hsien Liu[1,5¤]*

1 Division of Epidemiology, Department of Population and Quantitative Health Sciences, University of Massachusetts Medical School, Worcester, MA, United States of America, 2 Clinical and Population Health Research Program, Graduate School of Biomedical Sciences, University of Massachusetts Medical School, Worcester, MA, United States of America, 3 Graduate Medical Education, Internal Medicine, University of Massachusetts Medical School, Worcester, MA, United States of America, 4 Novartis Pharmaceuticals Corporation, East Hanover, NJ, United States of America, 5 Division of Rheumatology, Department of Medicine, University of Massachusetts Medical School, Worcester, MA, United States of America, 6 Division of Rheumatology, UMass Memorial Medical Center, Worcester, MA, United States of America

¤ Current address: University of Massachusetts Medical School, Worcester, MA, United States of America
* shaohsien.liu@umassmed.edu

**Data Availability Statement:** Data cannot be shared publicly because permission to do so was not obtained from the University of Massachusetts Medical School IRB or the participants. We also did

## Abstract

### Background

Many patients with axial spondylarthritis (axSpA) experience lengthy diagnostic delays upwards of 14 years. (5–14 years). Screening tools for axSpA have been proposed for use in primary care settings, but whether this approach could be implemented into busy primary care settings remains unknown.

### Objective

To solicit feedback from primary care physicians regarding questions from the Inflammatory Back Pain Assessment: the Assessment of Spondyloarthritis International Society (ASAS) Expert Criteria and gain insight about barriers and facilitators for implementing axSpA screening in primary care.

### Methods

Guided by Consolidated Criteria for reporting Qualitative Research (COREQ-criteria), we recorded, transcribed, and analyzed in-depth interviews with eight family medicine physicians and ten internists (purposeful sampling) using immersion/crystallization techniques.

### Results

Few physicians reported awareness of existing classification criteria for axSpA, and many reported a lack of confidence in their ability to distinguish between inflammatory and mechanical back pain. From three domains, 10 subthemes emerged: 1) typical work-up of axSpA patients in primary care, with subthemes including the clues involved in work-up and

not include the sharing of data in our informed consent procedures. The UMass Medical School IRB can be reached at IRB@UMassmed.edu (+1508 8564261).

**Funding:** Funding for this project was provided by Novartis Pharmaceuticals Corporation. Research reported in this publication was supported by the National Center for Advancing Translational Sciences of the National Institutes of Health under award number UL1TR000161 (KLL). This work was also supported by a charitable contribution to the UMass Memorial Foundation from Timothy S. and Elaine L. Peterson (JK, SHL). This project was also supported by the SAA/Jane Bruckel Early Career Investigator in AxSpA Award (SHL). The content is solely the responsibility of the authors and does not necessarily represent the official views of the NIH. The funders provided support in the form of salaries for authors [KLL, DS, SK, CD, JK, SHL, EY], but did not have any additional role in the study design, data collection and analysis, decision to publish, or preparation of the manuscript.

**Competing interests:** Dr. Kay has served as a consultant to pharmaceutical companies. Dr. Yi is an employee of Novartis. There are no patents, products in development or marketed products associated with this research to declare. These do not alter our adherence to PLOS ONE policies on sharing data and materials.

role of clinical examinations for axSpA; 2) feedback on questions from the Inflammatory Back Pain Assessment: ASAS Expert Criteria, with subthemes to evaluate contents/questions of a potential screening tool for axSpA; and 3) implementation of the screening tool in primary care settings, with subthemes of perceived barriers including awareness, time, other conditions to screen, rare disease, and lack of structured questionnaire for back pain and perceived facilitators including workflow issues and awareness.

## Conclusions

Primary care physicians believed that an improved screening instrument and a strong evidence-base to support the need for screening for axSpA are required. The implementation of axSpA screening into a busy primary care practice requires integration into the practice workflow, with use of technology suggested as a possible way to improve efficiency.

## Introduction

Axial spondyloarthritis (axSpA) is an inflammatory disease characterized by chronic back pain [1]. Prevalence estimates vary widely [2]. In the US, ~1% of the adult population is estimated to have axSpA [3], whereas in Canada the prevalence has been estimated to be 213 per 100,000 [4]. The diagnostic journey experienced by patients with axSpA is often lengthy, lasting upwards of 14 years in the United States [5–7], and delayed diagnosis can result in greater functional impairment, higher healthcare costs, and worse quality of life [8]. Early initiation of treatment can reduce symptoms [9], and may delay disease progression [9] and prevent disability [10].

Primary care providers have described reasons for delayed diagnosis of axSpA including disease characteristics (e.g., back pain is common, whereas axSpA is relatively uncommon); patient perception (e.g., sharing back pain at end of appointment); provider unfamiliarity (e.g., lack of awareness about axSpA); and healthcare system-related issues (e.g., brevity of primary care appointments) [11]. Although rheumatologists play a pivotal role in caring for patients with axSpA, most patients often consult with primary care providers (e.g., primary care physicians or internalist) seeking symptom relief when they initially have their symptoms (i.e., back pain or spinal pain) and are diagnosed by primary care providers other than rheumatologists [5]. Several sets of classification criteria exist for axSpA, including the modified New York criteria [12], the Berlin criteria [13], the Amor tool [14], the European Spondylarthropathy Study Group (ESSG) criteria [15], and the Assessment of Spondyloarthritis International Society (ASAS) criteria [16]. Yet, it is not known whether healthcare providers, other than those in rheumatology, are familiar with these criteria [17].

Several approaches to assist in identifying patients with axSpA have been explored in primary care settings, including the use of screening questionnaires [18, 19], early referral tools that combine clinical criteria and laboratory and imaging test results [20], and automated referral algorithms using electronic medical record (EMR) data [21]. However, few studies [22, 23] have engaged primary care providers to gather their opinions about screening tools for axSpA in clinical practice.

As part of a larger qualitative research study [**Sp**odylo**A**rthritis **S**creening and **E**arly **D**etection (SpA-SED) Study], the primary objective of this study was to evaluate primary care physician perspectives and views about the use of a screening tool for axSpA. This analysis focuses on statements made by primary care physicians regarding their baseline practices on typical

work-up of axSpA patients, feedback on questions from the Inflammatory Back Pain Assessment: the Assessment of Spondyloarthritis International Society (ASAS) Expert Criteria [19], and barriers and facilitators for implementation of axSpA screening in the primary care setting. The results of this study will provide foundational knowledge for the development and implementation of screening tools for axSpA.

## Materials and methods

The University of Massachusetts Medical School Institutional Review Board approved this study. All participants provided written informed consent.

### Study sample and setting

We recruited 18 primary care physicians to achieve saturation, with gender (8 women) and specialty balance (8 Family Medicine and 10 Internal Medicine) from January to May 2019. To be eligible, a physician must be a primary care physician in clinical practice and were able and willing to participate in an interview. Physicians were ineligible if they were: 1) unable to participate in a discussion lasting no more than 60 minutes; 2) unwilling to be audio recorded; 3) unable to consent; and 4) non-English-speaking subjects. Using purposeful sampling, we reached out to primary care physicians in Massachusetts and Rhode Island who were known to the authors and to regional professional societies (e.g., Rhode Island Academy of Family Physicians) to identify potential participants [24]. We identified 34 potential participants and completed 18 60-minute, audio-recorded interviews (3 in person, 15 on the phone, conducted by either KL, DS with SL or DS taking notes, no other people were present). No repeat interviews were conducted. One participant had conducted research (unrelated to axSpA) with the interviewer >5 years previously. Participants were compensated for their time with a $300 cash card.

### Study protocol

A multidisciplinary team developed the interview protocol guided by the consolidated criteria for reporting of qualitative research (COREQ) [25] and informed by a literature review and insights from the research team, which included two rheumatologists (S1 Table). We collected the qualitative data using in-depth interviews with a semi-structured interview outline. Each interview was conducted on a one-to-one basis with an observer from the research team taking notes. We began the interview with physicians' baseline practices on typical work-up of axSpA patients. We then solicited feedback during a question-by-question review of the Inflammatory Back Pain Assessment: ASAS Expert Criteria to understand their perspectives on those questions to screen axSpA patients [19]. The interviews were then completed by asking questions about implementation of axSpA screening tools in primary care settings. To understand the characteristics of physician participants in the study, we lastly administered a structured questionnaire at the end of interviews. It included items about socio-demographics, practice characteristics, awareness of the features of inflammatory back pain according to several different sets of classification criteria (i.e., Calin [26], Berlin [27], and ASAS [19]) and "any other criteria" (open-ended), and the following question: "How confident are you in distinguishing inflammatory back pain from mechanical back pain?" (extremely, very, somewhat, or not confident). Interviewers were members from the research team who had experience with in-depth interview and trained in qualitative research as well as for this study protocol. We also conducted pilot testing of the interview protocol with a primary care physician not participating in the study and did not include the data in this analysis. Interviewers reported no explicit bias

related to the topic under study but believed that primary care physicians may lack awareness about axSpA.

## Qualitative data analysis

We recorded each in-depth interview and discussion with the participants. We used a content analysis approach to completely transcribe and code the data collected. Transcription was carried out verbatim. We did not ask participants to review the transcripts of their interviews for comment or correction. We developed a hierarchical coding structure and handbook based on review of the first few transcripts and expanded throughout data analysis. Two research assistants performed line-by-line coding of the transcripts with NViVo qualitative software. Qualitative review of the dual coding confirmed that passages often appeared twice, suggestive of effective coding. The data were analyzed as a group using immersion / crystallization techniques [28]. We requested feedback from participants wiling to review results. This manuscript focuses on three themes in our coding structure: 1) baseline practices on typical work-up of axSpA patients; 2) feedback on the Inflammatory Back Pain Assessment: ASAS Expert Criteria questions; and 3) perspectives and views about implementation of axSpA screening tools in primary care settings.

## Results

On average, the study participants had been practicing in the field for a mean of 15.7 (±13.0) years (Standard Deviation) (Table 1). Most physicians were Non-Hispanic White, trained at US allopathic schools, and were affiliated with academic institutions. No family medicine physicians and only 30% of internal medicine physicians had heard of the ASAS classification criteria. No internists and 2 of 8 family medicine doctors reported feeling "extremely confident" and 3 of 8 family medicine and 3 of 10 of internists reported that they felt "very confident" in distinguishing inflammatory back pain from mechanical back pain.

In line with the interview protocols, 3 main themes along with 10 subthemes were established (Table 2).

### Typical baseline work-up of axSpA patients

When asking of the typical baseline work-up of axSpA patients (Table 3), physicians noted several clinical observations potentially pointing to axSpA, including: 1) young people presenting with back symptoms without any antecedent injury; 2) the presence of comorbid autoimmune conditions; 3) peripheral joint involvement, and other systemic manifestations (e.g., iritis). When speaking of the role of clinical examinations and tests, most physicians stressed that a thorough medical history would be essential for diagnosing axSpA. Some also expressed concern about having dissatisfied patients when costly tests are performed that may not yield definitive results.

### Feedback on questions from the Inflammatory Back Pain Assessment: ASAS Expert Criteria

With respect to questions in ASAS Expert Criteria to screen patients with AxSpA (Table 4), physicians were concerned that some questions were not specific (e.g., Have you suffered from back pain for more than 3 months?) or sensitive (e.g., Did your back pain develop gradually?). They were also concerned that patients with intermittent back pain might be missed. Physicians suggested including questions about decreased range of motion or stiffness, heel pain and other symptoms of enthesitis, psoriasis, and Crohn's disease. Most did not like the

**Table 1. Characteristics[a] of physician participants by specialty.**

| | Family Medicine (n = 8) | Internal Medicine (n = 10) |
|---|---|---|
| Age (years), mean (SD) | 52.9 (10.3) | 42.0 (12.7) |
| Women, % | 50.0 | 40.0 |
| Race/ethnicity, % | | |
| Non-Hispanic, White | 87.5 | 50.0 |
| Non-Hispanic, Black | 0 | 10.0 |
| Hispanic | 0 | 10.0 |
| Other | 12.5 | 30.0 |
| Trained at, % | | |
| US Allopathic school | 75.0 | 90.0 |
| US Osteopathic school | 0 | 0 |
| Foreign medical school | 25.0 | 10.0 |
| Years in practice, mean (SD) | 20.1 (13.4) | 12.6 (12.3) |
| Practice characteristics: (check all that apply), % | | |
| Individual | 0 | 0 |
| ≤ 5 physicians | 25.0 | 30.0 |
| ≥6 physicians | 62.5 | 50.0 |
| Hospital-based practice | 25.0 | 50.0 |
| Academic affiliation | 62.5 | 70.0 |
| Confidence in distinguishing inflammatory versus mechanical back pain, % | | |
| Not confident | 12.5 | 20.0 |
| Somewhat confident | 25.0 | 50.0 |
| Very confident | 37.5 | 30.0 |
| Extremely confident | 25.0 | 0.0 |
| Knowledge of inflammatory back pain classification criteria, % | | |
| Calin criteria | 0 | 0 |
| Assessment of Spondyloarthritis International Society criteria | 0 | 30.0 |
| Berlin criteria | 0 | 0 |

SD = Standard deviation; Percentages may exceed 100% due to rounding.

[a] For questions where respondents could select more than one answer choice, percentages may exceed 100%.

**Table 2. Developed main themes and subthemes.**

| Main themes | Subthemes |
|---|---|
| Typical baseline work-up of axSpA patients in primary care | Clues in working up patients |
| | Role of clinical examinations for axSpA |
| The Inflammatory Back Pain Assessment: the Assessment of Spondyloarthritis International Society (ASAS) Expert Criteria | Contents/questions for a potential screening tool for axSpA |
| Implementation of the screening tool in primary care settings | Perceived barriers: |
| | • Awareness |
| | • Time |
| | • Other conditions to screen |
| | • Rare disease |
| | • Lack of structured questionnaire for back pain |
| | Perceived facilitators: |
| | • Workflow issues |
| | • Awareness |

**Table 3. Synthesis of physician work-up of axial spondyloarthritis.**

| Subthemes | Synthesis | Representative quotes |
|---|---|---|
| **Clues in working up pointing to axSpA** | • Young age without any antecedent injury<br>• Comorbid conditions (e.g., psoriasis, Crohn's, other autoimmune conditions)<br>• Joint involvement in addition to back pain and other systemic involvement (e.g., eye conditions, peripheral joint involvement, "oddly shaped" digits)<br>• Decreased range of motion | • D17: The x-ray was suggestive, but I think it was also like a younger person without any, like, real good reason to have back pain. No, like, sports history, like, trauma or, you know, anything to really set it off.<br>• D12: If I'm getting that kind of positive history, I would probably go deeper to rule out other associated conditions, like uveitis, other joint pains, like in their hip, knees. 'Cause AS may have an SI joint problem, so they may say my—you know, my butt hurts and something. They may have all these digits maybe oddly shaped.<br>• D10: Certainly if they have a known condition like psoriasis or they could have psoriatic arthritis or rheumatoid arthritis or any other autoimmune kind of disease. If they're hunched over, you know, we'd think of ankylosing spondylitis. |
| **Role of clinical examinations for axSpA** | • Physicians stressed that a thorough medical history was essential to diagnosing axSpA<br>• Some physicians noted the importance of asking about family history<br>• Conducting a good physical examination is important. Are they hunched over? Can they bend over and touch toes?<br>• Physicians reported ordering the following images and lab tests:<br>• Plain X-rays, MRI (to look for bamboo spine)<br>• C-reactive protein, ESR, ANA, HLA-B27 | • D17:Well, I think it's important to take a good history. Because I think you're more likely to pick up on the symptoms suggestive of it with a thorough history and the family history as well.<br>• D5: There's actually a test where you can try to bend them over and measure the distance between standing between four vertebrae standing and flexing. So, most people can flex but if—ankylosing spondylitis, if it's all fused, you won't be able to do that. And so—anyway, so sometimes you can tell by physical exam.<br>• D23: imaging will show you—often they say, like, in the x-rays, it's like bamboo, it looks like bamboo along the spine. And so typically if I were to do imaging that's more what I would see. |

suggestion to immediately refer patients with 4 or 5 affirmative answers to a rheumatologist because of concerns about the shortage of rheumatologists; however, they thought that it would be helpful for specific tests to be suggested as next steps to pursue in evaluating these patients.

### Implementation of the screening tool in primary care settings

Physicians considered axSpA to be "uncommon" and perceived this as a barrier to implementation of a screening tool (Table 5). They suggested using such an instrument only in specific clinics (e.g., pain clinics), if a patient presented with recurrent back pain, or before ordering "expensive" tests. All physicians discussed lack of time as a barrier and pointed out that there are many other common conditions for which they need to screen. A few indicated that they would never use such a screening tool.

Table 6 summarizes primary care physicians' perceptions of facilitators to implementation of a screening tool for axial spondyloarthritis in primary care. Physicians expressed concern that patients using an online screening tool might request unnecessary referrals. Suggestions included administering the screening tool in the waiting room or by telephone before the visit, during the appointment when axSpA is suspected, or by using a clickable link in the EMR (Table 6). Physicians also suggested using machine learning to trigger when to initiate screening, embedding a screening tool in UpToDate (an online resource designed to provide physicians access to current clinical information), and using smart phrases in EMR software to record responses to the screening tool. Physicians want evidence to support the use of an axSpA screening tool in practice (e.g., sensitivity and specificity; US Preventive Services Task Force endorsement).

### Discussion

In our study, many primary care physicians lacked awareness of classification criteria for axSpA and most were not "extremely confident" in their ability to distinguish inflammatory

**Table 4. Synthesis of feedback from primary care physicians on Assessment of Spondyloarthritis International Society screening questions.**

| Question | Major comments | Representative quotes |
|---|---|---|
| **Have you suffered from back pain for more than 3 months?** | • Thought it was a reasonable question because mechanical back pain typically gets better in 6 weeks<br>• Thought the question lacked specificity<br>• Recognized that patients may experience flare-ups or intermittent pain and misinterpret the question | • D32: Yeah, I think that's in general we're taught six to 12 weeks is kind of what people should get, kind of between that acute and more chronic phase, so I think three months is an appropriate time.<br>• D7: One is the lack of specificity in the question, you know, because there's some people who, you know, will say I've had back pain for ten years and that means it's come and gone. And so whether that's a follow-up question or whether you put in the preface, you know, continued back pain or chronic back pain or back pain that hasn't gone away for the last three months. |
| **Did your back pain start when you were aged 40 or under?** | • Agreed that age is clinically relevant<br>• Thought that this question would be easy for patients to understand and answer<br>• Others thought that the question was "non-specific", "too broad", and "sensitive not specific"<br>• Questioned the value of this information because many people experience back pain under the age of 40 years and there are other contributors to pain (e.g., overweight) | • D31: So, I do like the one about under the age of 40 because I think that really fulfills the criteria that we often see with inflammatory spondyloarthropathies<br>• D25: It's a mix. I mean, the reason being—does that mean you had an episode of low back pain before you were 40? Well, 80% of the population has it during their lifetime and quite—you know, an enormous number under the age of 40. So, it's—I'd say it's a not particularly—it would—I would, you know, let's see, I don't know the properties of the test but it's not very specific. |
| **Did your back pain develop gradually?** | • Agreed this question likely distinguishes inflammatory back pain from pain resulting from an injury<br>• Thought most patients would be able to answer (yes or no).<br>• Thought the question was not sensitive enough<br>• Concerned that patients may not know when the back pain began, that patients would have recall bias, and that "gradually" was too vague and would be open to different interpretations<br>• Some suggested asking "Was there a specific incident you remember that caused your back pain?" which would have helped to bring to light an identifiable incident | • D32: I feel like that would be kind of a vague question, 'cause people are—kind of have different thoughts of how long gradually means. Like did it occur over a day, did it occur over a month, it's tough to say.<br>• D27: That makes sense, too, 'cause it distinguishes like an injury and like a—you know, it would have been a sudden start if it was an injury. |
| **Does your back pain improve with exercise?** | • Agreed that with inflammatory issues, improvement with exercise expected<br>• Expressed concern question was not sensitive or specific<br>• Did not like the term "exercise" as they felt the term was generic and vague. Not all patients "exercise" (because "exercise is in the eye of the beholder"); could be interpreted as "going to the gym"<br>• Concerned question may make patients feel "guilty and bad"<br>• Suggested alternative wording: "activity" or "move around" | • D34:'Cause exercise connotates I go to the gym and put on gym shorts and I pump iron. That's how I might have—that's what I think people might perceive, and it makes them feel guilty and bad, when you ask, "Do you exercise?" and they say, "No." They're reluctant to answer that. "Are you physically active?" "Yeah, I love going outdoors and playing with the kids, and we bike around." I get a lot more out of, "Are you physically active," versus, "Do you exercise?" I actually don't like the word "exercise." |
| **Do you find there is no improvement in your back pain when you rest?** | • Thought the question was good because it is something doctors don't think about<br>• Like that it gets at mechanisms of symptoms (e.g., stiffness)<br>• Thought that the "double negative wording" may be difficult for patients to answer<br>• Some thought this question could be combined with the question on exercise<br>• Suggested rewording: "When you rest, does your back get better?" | • D28: If they find there is no improvement in back when you rest, okay, that's something I haven't thought about, so I guess would be a good one.<br>• D29: It may make more sense, like, "Does your back pain improve with rest?" Because that's, like, not a negative question, right? |
| **Do you suffer from back pain at night which improves upon getting up?** | • Thought question could be answered easily by patients<br>• Addresses something doctors don't think about<br>• Ruled out osteoarthritis and was associated with morning stiffness<br>• Thought the question needed a more specific time frame: "better after 30 minutes into your day"; "more in the morning, better towards to evening"<br>• Thought the question was not specific or sensitive. They thought that chronic low back pain can be attributed to many things (e.g., disks when lying flat, bad mattress) | • D10: If that's suggestive of inflammatory spondylitis then I guess it's a reasonable question but I have a number of patients who say their back pain is worse first thing in the morning when they get up out of bed because they have a bad mattress,<br>• D27: Maybe thinking about, like—you could think about, like, worse when you first get up in the morning or better after 30 minutes into your day, something like that.<br>• D17: I think the last one is helpful, too, because it's more of a, like, morning stiffness. |

**Table 5. Primary care physician perceptions of perceived barriers to implement the screening tool for axial spondyloarthritis.**

| Subtheme | Representative quotes |
|---|---|
| Awareness | **D07:** Awareness is probably the biggest thing. I'm not aware of guidelines for screening, using a specific screening tool. If you don't know that it exists and it's evidence-based, that's probably the biggest. |
| Time | **D10:** Time, time... a primary care visit now involves . . .15 minutes or 30 minutes long and in those 15 minutes. . . we have to screen . . .for depression, . . .physical abuse at home, do their vital signs, and if . . . blood pressure is elevated, do it several times, get their medication list cleaned up, . . . completely reconcile the outside information now coming in through health information exchanges, talk to the patient, oh, by the way, examine them, come up with a plan, and document it all. |
| | **D22:** So, we have 20-minute patient encounters. Usually it takes five minutes to room a patient, then my time with the patient is from ten to 15 minutes, at the most, and then we have to go out and talk and do everything else, so time is definitely a constraint. I wish I had 40 minutes visits with the patient, but that's not possible. |
| | **D23:** Anything that takes extra time, primary care doctors do not have, and we already have so many forms and like notes–. . . There's just so much that's already expected of us as primary care doctors and so anything . . . that you can . . .prepopulate into the note and it's . . . yes or no questions,. . . . the easier and more efficient the better and the more likely people will use it. If it's cumbersome, people are less likely to integrate it in. |
| Other conditions to screen | **D12:** My clinic is not just a back clinic. There are so many other things. |
| | **D28:** I have a patient with back pain, they usually have four other things and then they say by the way, I have back pain. . . . it's overwhelming to do all that in 15 minutes. That's what we have; 15 minutes. |
| Rare disease | **D07:** . . . thinking about it. . . . thinking about these diagnoses is probably the biggest barrier. |
| | **D25:** Ankylosing spondylitis is rare and so . . .what we need public health campaigns about are, . . .clean water, clean air, exercise, healthy diet, safe sex, you know, birth control, domestic violence. . . .to have a public health campaign about a rare disease is. . . if there's something we can do about it significantly and change the course of the disease then it's kind of worthwhile but for symptom control, probably not. |
| Lack of structured questionnaire for back pain | **D10:** I don't have like a structured questionnaire or a structured note template for back pain. |

from mechanical back pain. Many of the physicians interviewed emphasized the importance of taking a good medical history. Primary care physicians indicated that they would value a screening tool that provided guidance on appropriate tests to order prior to a referral, rather than an immediate referral to a rheumatologist, because of the difficulty in obtaining a timely rheumatology consultation appointment. Barriers believed to impede implementation of screening for axSpA in primary care included lack of time (because other more common conditions require screening) and lack of awareness about axSpA. All physicians interviewed emphasized the importance of integrating any proposed screening strategy into their clinical workflow.

Physicians are concerned about overuse of testing and low-value healthcare given that dissatisfied patients when costly tests are performed that may not yield definitive results [29]. Primary care physicians have difficulty discriminating inflammatory back pain from other kinds of back pain and are often unaware of other SpA features that are important for the differential diagnosis. Add to this fact that the radiological proof is sometimes a late feature of the disease and the result is the diagnostic delay. As such, primary care providers in our study thought that an axSpA screening tool may be helpful if it could make use of visit time and health care

**Table 6. Primary care physician perceptions of perceived facilitators to implement the screening tool for axial spondyloarthritis.**

| Subtheme | Key areas | Representative quotes |
|---|---|---|
| Workflow issues | Simple and less time-consuming | D23: If there was a really easy website or smart phrase that I could get to, like . . .all the medical records systems they have. It's like Epic Care Connect, there are these things called smart phrases, they're like dot phrases. So if I were to say dot AS, . . .it would prepopulate all the, like, screening questions I would need to ask. |
| | Ease and efficiency/ maybe an app or algorithm/ incorporated into an EMR | D23: I think the biggest thing in terms of getting people to screen more is doing something that is very efficient and easy to implement. |
| | | D01: Maybe some sort of form for the patient or an app, something where they could fill out so that . . . the questions would be completed before I start interviewing them and I could just review the questions with them. . . |
| | | D31: doing something that would make it more facilitated to . . .incorporate the criteria into an electronic medical record. I think that's what will make like easier for everybody, where I wouldn't have to pull out my phone, where all I would do is . . .click something or click a pulldown menu and say yes, yes, no, yes, you know, that kind of thing. |
| | Use of templates and health information technology | D07: I do think that the idea of how to integrate it, especially in busy primary care centers, is tough . . .I'm not sure that I have the perfect idea because on the one hand if you're just considering this and just thinking about back pain, . . .I think it all makes perfect sense, but yet creating the perfect template, . . .this is back pain, but then there's five other things, so this idea of using this template sounds good when there's only one thing to pay attention to, so that in some ways is an urgent care model. |
| | Assistance from team | D01: Certainly in my clinical flow, we could definitely address the time constraints, like having a staff person ask these questions when somebody has a chief complaint of back pain and . . .it's new. . . |
| | | D27: A standardized approach. . . .we have a ton of templated things that our medical assistants fill out at various frequencies with our patients before the physician or NP enters the room, so it wouldn't be difficult to implement. |
| Awareness | Education | D22: . . .we just need to realize that. . .this is one possibility of pain that is there, so we first of all have to keep this in mind as one of the differential, sometimes we don't. |
| | | D24: . . .I think awareness to this is important and . . .any of those screening tools would be good and, again, . . .being family medicine, most of what I get is through. . .one of the journals that we use. |
| | UpToDate/ journals/ conferences | D22: I would say for 90% of my patients that I'm not really sure what's going on with them . . .before I go to see the patient, I almost always open UpToDate. For example, the patient is coming in with something unusual that I haven't seen before, I would just go to UpToDate, go to the summary and recommendation center part of UpToDate and just read that, a little bit about symptoms, a little bit about treatment. |
| | Data driven | D17: Well, I think evidence is always helpful . . ..data supporting that incorporating a specific question set, screening questions, into your practice improves outcomes in some way, or increases diagnosis of these inflammatory causes of back pain. I think that kind of stuff is convincing to people. It's always nice to have data to support your clinical practices, and I think . . .it's just kind of a reminder when you see something pop in Journal Watch or something, . . .that you should be incorporating it, and you should adjust your practice if you're not already using a tool like this and one that is data driven. |
| | US Preventive Task Force | D31: And that screening guideline is going to be most likely best employed if you do it through the United States Preventative Service Taskforce. |

resources more efficient. Pressure from national campaigns to reduce ordering of imaging (e.g., Choosing Wisely [30]), patient concerns about the high costs of laboratory tests that are not fully covered by insurance [31], and the workforce shortage in rheumatology [32] all underscore the value of using screening tools to identify patients in need of appropriate additional testing and referral to evaluate for axSpA.

Given the brief amount of time available to spend with patients in primary care appointments [33], physicians reinforced the notion that implementation of an axSpA screening tool must not require additional physician time. Because the prevalence of axSpA in the general population is low, a screening approach would need to target patients with chronic back pain and rely on patient to complete a questionnaire. Participants suggested working through the

US Preventive Task Force before making recommendations for screening in primary care practices [34]. This process begins with nominating a topic. If selected for further consideration, the process would involve development of a research plan and review of published evidence, culminating in a final recommendation statement.

In our study, physicians remarked that back pain tends to be a 'doorknob' question–one that is left to the end of the visit when time already has run out. Short appointment times impede a physician's ability to fully explore the underlying reasons for back pain [30]. Patients experiencing waxing and waning symptoms may be misidentified as experiencing new onset back pain.

Participants in our study recommended various approaches using technology to assist with axSpA screening. Incorporating axSpA screening questions into EMRs has been suggested as being useful to improve efficiency in physicians' offices [35]. Such an approach has encouraged physician adherence to following recommended screening guidelines and has been successful in some [36, 37], but not all, case studies [38]. Approaches that engage non-clinician support staff [39] or employ automated tools improved screening in primary care settings [40].

Patient reported screening approaches (e.g., via online screening or an app) offer an alternative to EMR-based algorithms. Use of a non-invasive, algorithm to find cases of axSpA, based on information contained in the EMR, has been studied using a randomized controlled cluster trial design [41]. In the intervention arm, the diagnosis of axSpA was ultimately confirmed in 8% of subjects but this approach did not have a short term impact on physical function, as average disability scores were similar in both arms at 4 months [41].

## Strengths/Limitations

The qualitative study design is a strength, as this approach often provides insights that cannot be obtained by asking more closed-ended quantitative research questions. Our protocol adhered to best practices for qualitative research. Despite that a convenience sample was used [42] and some participants recruited were known to the researchers practicing medicine in Massachusetts and Rhode Island, the primary interviewer (KLL) was not someone with a prior relationship with the participants and therefore the interpretation bias could be reduced. Despite that none of the physicians recalled ever having diagnosed a patient with axSpA, most were confident in distinguishing inflammatory versus mechanical back pain. In addition, the average years in practice was 15.7 years and the majority were affiliated with academic institutions. Although primary care practices in the Northeast may differ from those in other regions of the United States, the findings do not appear to be overly optimistic with respect to the approach of screening for early detection of axSpA. None recalled ever having diagnosed a patient with axSpA, although several reported having treated patients with ankylosing spondylitis that was diagnosed before coming under their care.

## Conclusions

Primary care physicians believed that the delay in diagnosis of axSpA is too long. With respect to the ASAS screening questions, physicians agreed that these questions need improvement and noted that some questions were neither sensitive nor specific. Primary care physicians preferred a screening tool that recommends additional testing, rather than one that directs referral to a rheumatologist. They believed that there may be a role for use of such a screening tool in the primary care setting but requested evidence to support its implementation, since they already must follow many other recommendations to screen for conditions more common than axSpA. Strategies to implement axSpA screening must be mindful of practice workflow issues and should be effective in reducing delay in diagnosis of axSpA.

## Supporting information

**S1 Checklist. COREQ (COnsolidated criteria for REporting Qualitative research) check-list.**
(PDF)

**S1 Table. Physician interview questions.**
(DOCX)

## Acknowledgments

We thank the primary care physicians who participated in this study. We also thank Jina Park and Katarina Ferrucci for their assistance with this project.

## Author Contributions

**Conceptualization:** Kate L. Lapane, Catherine Dube, Shao-Hsien Liu.

**Data curation:** Kate L. Lapane, Divya Shridharmurthy, Sara Khan, Ariel Beccia.

**Formal analysis:** Kate L. Lapane, Divya Shridharmurthy, Sara Khan, Ariel Beccia.

**Funding acquisition:** Kate L. Lapane, Esther Yi, Jonathan Kay, Shao-Hsien Liu.

**Investigation:** Kate L. Lapane, Esther Yi, Jonathan Kay, Catherine Dube, Shao-Hsien Liu.

**Methodology:** Kate L. Lapane, Divya Shridharmurthy, Ariel Beccia, Catherine Dube, Shao-Hsien Liu.

**Project administration:** Divya Shridharmurthy, Sara Khan, Ariel Beccia, Shao-Hsien Liu.

**Resources:** Esther Yi.

**Writing – original draft:** Kate L. Lapane, Sara Khan, Daniel Lindstrom, Shao-Hsien Liu.

**Writing – review & editing:** Kate L. Lapane, Divya Shridharmurthy, Sara Khan, Daniel Lindstrom, Ariel Beccia, Esther Yi, Jonathan Kay, Catherine Dube, Shao-Hsien Liu.

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
