## [Decision Letter · Decision Letter 0]

20 Jan 2021

PONE-D-20-26736

Primary care physician perspectives on screening for axial spondyloarthritis: A qualitative study

PLOS ONE

Dear Dr. Liu,

Thank you for submitting your manuscript to PLOS ONE. After careful consideration, we feel that it has merit but does not fully meet PLOS ONE’s publication criteria as it currently stands. Therefore, we invite you to submit a revised version of the manuscript that addresses the points raised during the review process.

Please address the comments raised by the four reviewers.

We look forward to receiving your revised manuscript.

Kind regards,

Alison Rushton

Academic Editor

PLOS ONE

Journal Requirements:

2. Please provide additional details regarding participant consent. In the ethics statement in the Methods and online submission information, please ensure that you have specified (1) whether consent was informed and (2) what type you obtained (for instance, written or verbal, and if verbal, how it was documented and witnessed). If the need for consent was waived by the ethics committee, please include this information.

3. In your Methods section, please provide additional information about the participant recruitment method and the demographic details of your participants. Please ensure you have provided sufficient details to replicate the analyses such as: a) the recruitment date range (month and year), and b) a description of any inclusion/exclusion criteria that were applied to participant recruitment.

4. Thank you for stating the following in the Financial Disclosure section:

"Funding for this project was provided by Novartis Pharmaceuticals Corporation. Research reported in this publication was supported by the National Center for Advancing Translational Sciences of the National Institutes of Health under award number UL1TR000161 (KLL). This work was also supported by a charitable contribution to the UMass Memorial Foundation from Timothy S. and Elaine L. Peterson (JK, SHL). This project was also supported by the SAA/Jane Bruckel Early Career Investigator in AxSpA Award (SHL). The content is solely the responsibility of the authors and does not necessarily represent the official views of the NIH. The funders had no role in study design, data collection and analysis, decision to publish, or preparation of the manuscript."

We note that one or more of the authors have an affiliation to the commercial funders of this research study : Novartis Pharmaceuticals Corporation.

4.1. Please provide an amended Funding Statement declaring this commercial affiliation, as well as a statement regarding the Role of Funders in your study. If the funding organization did not play a role in the study design, data collection and analysis, decision to publish, or preparation of the manuscript and only provided financial support in the form of authors' salaries and/or research materials, please review your statements relating to the author contributions, and ensure you have specifically and accurately indicated the role(s) that these authors had in your study. You can update author roles in the Author Contributions section of the online submission form.

4.2. Please also provide an updated Competing Interests Statement declaring this commercial affiliation along with any other relevant declarations relating to employment, consultancy, patents, products in development, or marketed products, etc.  

Reviewers' comments:

Reviewer's Responses to Questions

**Comments to the Author**

1. Is the manuscript technically sound, and do the data support the conclusions?

Reviewer #1: Partly

Reviewer #2: Yes

Reviewer #3: Yes

Reviewer #4: No

2. Has the statistical analysis been performed appropriately and rigorously? 

Reviewer #1: N/A

Reviewer #2: Yes

Reviewer #3: Yes

Reviewer #4: N/A

3. Have the authors made all data underlying the findings in their manuscript fully available?

Reviewer #1: No

Reviewer #2: Yes

Reviewer #3: Yes

Reviewer #4: No

4. Is the manuscript presented in an intelligible fashion and written in standard English?

Reviewer #1: No

Reviewer #2: Yes

Reviewer #3: Yes

Reviewer #4: Yes

5. Review Comments to the Author

Reviewer #1: 

I thank the authors the work they did and method they used to respond to their research question. However, their work need to be reviewed before it is published. Below are some observations and comments:

1. Line 60-61: In the background you mention that the diagnostic delay is 5 to 14 years in while in the introduction of the manuscript you mention 5 years or longer. I expected to see the same in the manuscript, which I did not. Moreover, cross check your reference. It is not clear to me how a published manuscript on the same study with the same authors published very recently mentions 5 to 10, and in this study, you mention 5 to 14. Just to double check if it is in line with the used references or there may be any inaccuracy.

2. Line 69: “the modified New York” sounds incomplete; add the word that may be missing; criteria or tool??

3. Why did you purposively recruited primary health care physicians and internal medicine specialists? What was behind that purpose? Can you add something on setting sections about how the system works? Just for a reader who is unfamiliar with US health system.

4. Minimize the use of the passive voice wherever it is possible.

5. Why didn’t you keep this question qualitative? “How confident are you in distinguishing inflammatory back pain from mechanical back pain?”

6. Table 5 and 6 show who said what, that is not done for tables 3, and 4. It would be better if you keep it consistent. It also looks like the participants’ language on main themes 1 and 2 is missing. In qualitative studies, I expect to read partipants’ own words instead of rephrased sentences. Keep the same style for all themes.

7. I appreciate the introductions to each subtheme. However, it would be better if you do not separate them from the exerpts you put in the tables. Using a narrative style as if you are telling your story makes it easier for the reader. You do not have to change if you prefer that structure.

8. Please elaborate more on the first theme in both result and discussion sections.

Reviewer #2: 

This is a research to answer a relevant question to primary healthcare providers. The choice of the study design was good and the analysis of the data is adequate.

Some bias exist inevitably, but this was highlighted well in the discussion section. I do not think that this affect the quality of the research or the validity of the results.

Reviewer #3: 

The am this study to solicit feedback from primary care physicians regarding questions from the Inflammatory Back Pain Assessment: the Assessment of Spondyloarthritis International Society (ASAS) Expert Criteriaand gain insight aboutbarriers and facilitators for implementingaxSpA screening in primary care. This is a very important study.

Reviewer #4: 

There article is nicely written.

The authors have presented a qualitative research (study instrument being in-depth interviews with 18 primary care physicians) to explore primary care physicians' perspective on screening for axial spondyloarthritis.

There are following observations/concerns:

1. Most of the physicians recruited for the study were known to the authors. Considering nature of the study, this could have introduced interpretation bias among the recruited physician(s).

2. None of the physicians recalled ever having diagnosed a patient with axial spondyloarthritis.

6. PLOS authors have the option to publish the peer review history of their article (what does this mean?). If published, this will include your full peer review and any attached files.

Reviewer #1: No

Reviewer #2: No

Reviewer #3: No

Reviewer #4: No

---

## [Author Response · Author response to Decision Letter 0]

1 Apr 2021

Please see attached document for response to reviewers. Thank you!

---

## [Decision Letter · Decision Letter 1]

10 May 2021

Primary care physician perspectives on screening for axial spondyloarthritis: a qualitative study

PONE-D-20-26736R1

Dear Dr. Liu,

We’re pleased to inform you that your manuscript has been judged scientifically suitable for publication and will be formally accepted for publication once it meets all outstanding technical requirements.

Kind regards,

Alison Rushton

Academic Editor

PLOS ONE

Additional Editor Comments (optional):

Thank you for addressing the reviewers' feedback to a satisfactory level.

Reviewers' comments:

Reviewer's Responses to Questions

**Comments to the Author**

1. If the authors have adequately addressed your comments raised in a previous round of review and you feel that this manuscript is now acceptable for publication, you may indicate that here to bypass the “Comments to the Author” section, enter your conflict of interest statement in the “Confidential to Editor” section, and submit your "Accept" recommendation.

Reviewer #1: All comments have been addressed

Reviewer #3: All comments have been addressed

2. Is the manuscript technically sound, and do the data support the conclusions?

Reviewer #1: Yes

Reviewer #3: Yes

3. Has the statistical analysis been performed appropriately and rigorously? 

Reviewer #1: N/A

Reviewer #3: Yes

4. Have the authors made all data underlying the findings in their manuscript fully available?

Reviewer #1: No

Reviewer #3: Yes

5. Is the manuscript presented in an intelligible fashion and written in standard English?

Reviewer #1: Yes

Reviewer #3: Yes

6. Review Comments to the Author

Reviewer #1: Profuse thanks to the authors for minutely responding to each single comments and their zeal to go back to transcripts for more quotes. Their work looks great and ready to be published.

Reviewer #3: This study To solicit feedback from primary care physicians regarding questions from the Inflammatory Back Pain Assessment: the Assessment of Spondyloarthritis International Society (ASAS) Expert Criteria and gain insight about barriers and facilitators for implementing axSpA screening in primary care. This is a very important study with results important.

7. PLOS authors have the option to publish the peer review history of their article (what does this mean?). If published, this will include your full peer review and any attached files.

Reviewer #1: No

Reviewer #3: No

---

## [Editor Report · Acceptance letter]

14 May 2021

PONE-D-20-26736R1 

Primary care physician perspectives on screening for axial spondyloarthritis: a qualitative study 

Dear Dr. Liu:

I'm pleased to inform you that your manuscript has been deemed suitable for publication in PLOS ONE. Congratulations! Your manuscript is now with our production department. 

Kind regards, 

on behalf of

Professor Alison Rushton 

Academic Editor

PLOS ONE